# Fractional Flow Reserve-Guided Coronary Revascularization: Evidence from Randomized and Non-Randomized Studies

**DOI:** 10.3390/diagnostics12112659

**Published:** 2022-11-01

**Authors:** Luca Paolucci, Fabio Mangiacapra, Michele Mattia Viscusi, Annunziata Nusca, Giuseppe Zimbardo, Pio Cialdella, Michael Edward Donahue, Leonardo Calò, Gian Paolo Ussia, Francesco Grigioni

**Affiliations:** 1Unit of Cardiovascular Science, Department of Medicine, Campus Bio-Medico University, 00128 Rome, Italy; 2Division of Cardiology, Policlinic Casilino, 00169 Rome, Italy

**Keywords:** coronary artery disease, coronary revascularization, fractional flow reserve

## Abstract

Simple visual estimation of coronary angiography is limited by several factors that can hinder the proper classification of coronary lesions. Fractional flow reserve (FFR) is the most widely used tool to perform a physiological evaluation of coronary stenoses. Compared to isolated angiography, FFR has been demonstrated to be more effective in selecting those lesions associated with myocardial ischemia and, accordingly, impaired outcomes. At the same time, deferring coronary intervention in those lesions that do not show ischemic FFR values has proven safe and not associated with adverse events. Despite a major randomized clinical trial (RCT) and several non-randomized studies showing that FFR-guided revascularization could be superior to isolated angiography in improving clinical outcomes, subsequent RCTs have reported conflicting results. In this review, we summarize the principles behind FFR and the data currently available in the literature, highlighting the main differences between randomized and non-randomized studies that investigated this topic.

## 1. Introduction

Proper classification of coronary lesions (associated with myocardial ischemia or not) is crucial in order to guide coronary revascularization and avoid useless and potentially dangerous procedures. Fractional flow reserve (FFR) is the most commonly used method to perform invasive physiological evaluations of coronary stenoses [1] and effectively identify those lesions needing percutaneous coronary intervention (PCI) or coronary artery bypass grafting (CABG). Unlike FFR or other physiological indexes, simple visual estimation of coronary angiography is limited by several factors, including inter/intra-observers variability, oculo-stenotic reflex, epicardial spasm, vessels foreshortening or overlap, and others [2]. On the other hand, FFR is an objective and reproducible estimation of myocardial ischemia and is not limited by most of these conditions. Notably, it has been demonstrated that there is a high percentage of discordance between the treatment strategy chosen by operators according to isolated visual estimations and the one that would have been adopted according to the FFR values, indicating a significant tendency toward the overtreatment of coronary lesions [3]. Since its first introduction, FFR utilization has progressively increased during recent years [4]. According to European and American recommendations, FFR should be used in patients without documented evidence of myocardial ischemia and/or patients with intermediate coronary lesions [5,6]. After the positive results produced in the one-year data of the “Fractional Flow Reserve Versus Angiography for Multivessel Evaluation” (FAME) study [7], several randomized clinical trials (RCTs) [8,9,10,11,12,13,14,15,16,17] and non-randomized [18,19,20,21,22,23,24,25,26,27,28,29,30,31] studies investigated the effectiveness of FFR in improving clinical outcomes. Overall, despite data from non-randomized studies strongly supporting the benefit of an FFR-guided revascularization strategy compared to isolated angiography in terms of the clinical endpoint, several RCTs have reported conflicting results. The purposes of this narrative review are as follows: (1) to summarize the rationale behind the potential benefits of FFR-guided coronary revascularization; (2) to briefly report the evidence and results produced by the largest observational studies and RCTs comparing FFR to isolated angiography; (3) to critically propose possible explanations for the conflicting data in the available literature.

## 2. Physiological Principles of FFR

FFR was first described and successively tested in human subjects by Pijls and colleagues in 1993 [32] and 1996 [33]. Conceptually, FFR is a ratio between two flows: one through a coronary artery ideally positioned beyond a stenotic segment (*Q*), and one through an ideally identical coronary artery without any stenotic segments (*Qn*). The ratio between *Q* and *Qn* is considered an index of myocardial ischemia. According to Ohm’s law when applied to fluid dynamics, coronary and myocardial flow can be described as a ratio between Δ*P* and *R*, where Δ*P* is the pressure gradient across the artery and R the sum of epicardial and microvascular resistances. When calculating *Qn*, ΔP represents the difference between aortic mean pressure (*Pa*) and central venous pressure (*Pv*). Similarly, when calculating *Q*, Δ*P* is the difference between distal coronary mean pressure (*Pd*) and *Pv*. In this latter case, *Pd* represents an equivalent of *Pa* in an ideal vessel sited beyond the stenotic segment. Therefore, the FFR formula can be reported as follows:(1)$$\mathrm{FFR}:\;\frac{Q}{Qn}=\frac{\frac{\left(Pd-Pv\right)}{R}}{\begin{array}{c} \frac{\left(Pa-Pv\right)}{R} \end{array}}$$

In this formula, *Pv* can be omitted due to its small value compared to *Pd* and *Pa*. Similarly, considering that we are evaluating the ratio between two flows derived from the same vessel (one ideal vessel without the stenosis and one ideal vessel sited beyond the stenosis), the values of *R* are the same and, therefore, can be eliminated. Following that logic, the derived FFR formula corresponds to:(2)$$\mathrm{FFR}:\;\frac{Q}{Qn}=\frac{Pd}{Pa}$$

The main assumption of Ohm’s formula is that the relationship between *Q*, Δ*P*, and *R* is linear. However, in vivo, coronary flows and pressures change during each cardiac cycle and according to several dynamic factors [34,35]. In order to linearize the relationship between *Q*, Δ*P*, and *R* and to minimize the influence of the microcirculatory resistance, maximal coronary hyperemia must be achieved by using adenosine or papaverine infusion [36,37]. Although their discussion is beyond the purposes of this review, it should be noted that several other indexes have been created to perform the physiological evaluation of coronary lesions without the need for inducing maximal hyperemia [38,39]. Most of them have been validated using FFR as the gold-standard reference. Initially, a threshold of FFR < 0.75 was proposed to define myocardial ischemia; however, further studies have demonstrated that higher values are more discriminative, and a threshold of FFR < 0.80 is currently used [7]. Although FFR has been traditionally considered an index of epicardial ischemia, it can be strongly influenced by the microvascular function and resistances, with a tendency toward an elevation of its values [40]. Nevertheless, in patients with microvascular dysfunction or myocardial infarction with non-obstructive coronary artery disease (MINOCA) and concomitant intermediate coronary lesions, FFR can help rule out type 1 MI epicardial ischemia [41,42].

## 3. The Relationship between FFR and Clinical Outcomes

One of the main principles of FFR-guided revascularization is that deferring coronary interventions in lesions showing non-ischemic FFR values is safe and not associated with impaired outcomes [43,44]. This concept has been confirmed even at long-term follow-ups, as demonstrated by the 15-year follow-up results from the DEFER study [45]. In this trial, 325 patients with intermediate coronary lesions were recruited. Patients showing non-ischemic FFR values (n = 181) were randomly assigned to deferral or performance of PCI. At a 15-year follow-up, no significant difference in death rate was evident between the two groups (33.0% vs. 31.1%, *p* = 0.79). Moreover, patients recruited in the deferral group showed significantly lower rates of MI (2.2% vs. 10.0%, *p* = 0.03), probably due to reduced rates of stent failure-related acute events. Together with the evidence showing that FFR-negative lesions do not necessitate coronary interventions and are not associated with an increased risk of adverse events, several studies have demonstrated that FFR-positive lesions can strongly influence clinical outcomes. A large meta-analysis by Johnson et al. demonstrated that major adverse cardiovascular events (MACE) are linearly associated with FFR values, with a progressive increase in the number of events for progressively lower values of FFR [46]. Notably, this work demonstrated that FFR should be considered a continuous variable inversely associated with clinical events rather than a categorical cut-off for ischemia. Therefore, ischemic lesions with very low FFR values tend to show worse outcomes compared to ischemic lesions with higher FFR values. In a study involving 1029 stable coronary lesions treated with medical therapy, Barbato et al. showed that lesions generating MACE had significantly lower values of FFR compared to those that did not (0.68 vs. 0.80, *p* < 0.001). In this work, FFR was an independent predictor of MACE at a two-year follow-up in a multivariate Cox’s model, including both clinical and angiographic variables [47]. Regarding the risk of future myocardial infarctions (MI), it has been proven that patients with FFR-positive lesions show higher levels of serum-oxidated low-density lipoprotein and secretory phospholipase A2, which may be associated with an increased risk of atherosclerotic plaque destabilization [48]. The association between FFR-positive lesions and MI has been further confirmed in a recent sub-study of the Compare Acute trial by Piròth et al. [49]. In this study involving patients with ST-elevation MI (STEMI) and multivessel disease, the authors demonstrated that non-infarct-related FFR-positive lesions that were not treated with a PCI had a higher risk of generating MIs during a two-year follow-up.

Following a PCI or CABG, several early or late complications can occur, potentially impairing prognosis. These complications include myocardial injury/procedural myocardial infarction [50,51,52], early/late stent thrombosis [53], stent or surgical graft failures [54,55], acute kidney injury [56], and others. Moreover, in patients undergoing a PCI, dual antiplatelet therapy (DAPT) administration following the procedure can be associated with minor and major bleeding [57]. These and other adverse events following coronary revascularization can strongly influence long-term clinical outcomes and have been advocated as possible contributors to the suboptimal results obtained in major RCTs when comparing PCIs or CABG with optimal medical therapy [58,59]. Overall, FFR-guided revascularization could potentially preserve the benefits of coronary interventions and reduce the risk of early and late complications, primarily by deferring useless procedures and directing the revascularization only to those lesions causing myocardial ischemia.

## 4. Evidence from Non-Randomized Studies

The effectiveness of FFR-guided revascularization compared to isolated angiography has been investigated in several registries and retrospective observational studies involving different clinical settings. Most of these studies demonstrated that FFR was associated with significantly improved clinical outcomes [19,22,25,60]. Despite a large proportion of these investigations being conducted immediately after the published FAME study, their findings have been further confirmed in recent large national registries. An analysis of the “Swedish Coronary Angiography and Angioplasty Registry” (SCAAR), including 23,860 patients undergoing a PCI due to stable coronary disease, showed that FFR-guided revascularization (n=3367) was associated with reduced rates of all-cause death, stent thrombosis, and restenosis at 4.7 years of median follow-up. Similar results have been reported in two large multicenter national registries from France and the United States, which both showed improved clinical outcomes in terms of MACE and all-cause death even after adjustment for several confounders [4,31]. Few studies have been conducted with the purpose of investigating non-common endpoints such as short-term mortality or in-hospital bleeding. Recently, Omran et al. retrospectively collected data from a registry of 304,548 patients with acute coronary syndromes (ACS) undergoing a PCI [29]. Despite the group of patients undergoing FFR-guided PCI being largely underrepresented in their study (n = 7832), the authors were able to demonstrate that this strategy led to reduced rates of in-hospital mortality, peri-procedural bleeding, acute kidney injury, and total length of stay. Despite coming from a retrospective analysis, these results suggest that an FFR-guided strategy could be effective in improving other endpoints different from the ones usually considered in RCTs, which can significantly influence the patient’s prognosis.

## 5. Evidence from RCTs

At present, 10 RCTs comparing FFR-guided revascularization to an angiography-guided one have been published, including 5583 patients (2804 in the FFR arms and 2779 in the angiography arms) [8,9,10,11,12,13,14,15,16,17]. However, these RCTs differ in terms of sample size, clinical settings, and revascularization modalities. The FAME study was the first RCT investigating this topic and recruited 1005 patients with multivessel disease who underwent an FFR-guided or angiography-guided PCI [7]. At a one-year follow-up, the FFR-guided strategy led to reduced rates of the composite endpoint of death, myocardial infarction and repeat revascularization (18.3% vs. 13.1%, *p* = 0.02). These findings were further confirmed at a two-year follow-up, when the composite endpoint appeared to be driven mainly by a reduction in other MI events in the FFR arm (9.9% vs. 6.1%, *p* = 0.03) [61]. Notably, the difference between the two strategies was not statistically significant when follow-up was prolonged to five years, despite persisting in absolute percentages (31% vs. 28%, *p* = 0.31) [11]. As the authors reported, the reduction of the sample size caused by loss of patients during follow-up could have potentially underpowered their five-year analysis. The “Proper Fractional Flow Reserve Criteria for Intermediate Lesions in the Era of Drug-eluting Stent” (DEFER-DES) trial randomized 229 patients with intermediate coronary lesions to a routine PCI or FFR-guided intervention. Despite FFR leading to a significant reduction in the number of patients undergoing a PCI, no benefit was evident over isolated angiography at a five-year follow-up [9]. It should be noted that the DEFER-DES trial used a different threshold to define myocardial ischemia (FFR < 0.75) and that it was prematurely interrupted due to concerns regarding the rates of late stent thrombosis. Similar findings were reported in the “Double Kissing Crush Versus Provisional Stenting Technique for Treatment of Coronary Bifurcation Lesions VI” (DK-CRUSH VI) trial, which showed that FFR was not associated with reduced rates of adverse events despite leading to fewer side-branch interventions in patients with true bifurcation lesions undergoing a PCI [8]. Two RCTs have been published investigating this topic in the setting of patients with non-STEMI (NSTEMI), both showing that FFR was effective in reducing the number of lesions and patients actually revascularized without improving clinical endpoints [10,12]. The same results were reported in two RCTs enrolling patients treated with CABG [13,14], which showed that FFR guidance was not associated with improved outcomes despite potentially leading to a reduced number of positioned surgical grafts. Compared to the FAME study, all the discussed RCTs were relatively small as they recruited fewer than 200 patients per group. However, during the last two years, three major RCTs with sample sizes similar to the one in the FAME study have been published. The “FLOW Evaluation to Guide Revascularization in Multi-vessel ST-elevation Myocardial Infarction” (FLOWER-MI) trial investigated whether an FFR-guided PCI of non-culprit lesions in patients with STEMI and multivessel disease was superior to standard angiography. In that study, which randomized 1163 patients, FFR failed in improving clinical outcomes compared to angiography (composite endpoint: 5.5% vs. 4.2%, *p* = 0.31). The most recently published studies include the “Functional Testing Underlying Coronary Revascularisation” (FUTURE) trial and the “In the Management of Coronary Artery Disease, Does Routine Pressure Wire Assessment at the Time of Coronary Angiography Affect Management Strategy, Hospital Costs, and Outcomes?” (RIPCORD 2) trial. Both failed in demonstrating any benefit associated with FFR-guided revascularization (composite endpoint in the FUTURE trial: 14.6% vs. 14.4%, *p* = 0.85; composite endpoint in the RIPCORD 2 trial: 9.5% vs. 8.7%, *p* = 0.64) [16,17]. Similar to the DEFER-DES trial, the FUTURE trial was prematurely interrupted due to safety concerns when the enrollment phase was near 50% of the planned sample size. The features and results of the reported RCTs have been summarized in Table 1.

## 6. Entry Bias across the RCTs

Despite several non-randomized observational studies [4,20,23,24,25,29,36] and data from the first two years of follow-up of the FAME trial [7,61] strongly supporting the superiority of FFR compared to isolated angiography, the results from the latest RCTs are challenging this hypothesis. Understanding the possible reasons underlying these discrepancies can be difficult. Non-randomized observational studies are exposed to a high risk of “selection bias”, which is related to the non-randomized nature of the treatment allocation [62]. This is evident when we observe data from observational studies derived from national registries, which show that patients undergoing FFR-guided revascularization are often younger, with lower rates of previous CABG, MI, and stroke, reduced left ventricular ejection fraction, left main disease, and ACS [22,31]. As showed by Parikh et al., the risk of FFR utilization in real-world patients may be reduced in the elderly and those with peripheral artery disease or previous heart failure. At the same time, it can be increased in those with multivessel disease, previous PCIs, and normal cardiac function [4]. Following this, several biases associated with patient selection could potentially influence the results of observational studies. In order to partially overcome this issue, many of the works investigating this topic have used propensity score-matching methods to simulate a randomized allocation [23,24,25,28]. Nevertheless, although they showed promising results, residual bias related to possible confounders cannot be ruled out [62]. Despite these major limitations, non-randomized observational studies often recruit real-world cohorts of patients that tend to represent more accurately those who physicians treat in clinical practice [63]. On the other hand, RCTs are exposed to the risk of “entry bias”, caused by the highly selective inclusion/exclusion criteria, which often lead to recruiting low-risk patients, poorly representative of a real-world population [63]. Most of the RCTs investigating FFR-guided revascularization consistently excluded patients with left main coronary disease, chronic total occlusions, STEMI, previous CABG, recent MI, or severely calcified coronary disease [7,12,15,16,17].

## 7. Sample Sizes, Clinical Events Rates and Heterogeneity across the RCTs

Overall, despite 10 RCTs having been conducted, only four of them included more than 400 patients (FAME, FLOWER MI, FUTURE, and RIPCORD 2) [7,15,16,17], and three were early interrupted due to safety concerns or slow recruitment [9,14,16]. Regarding the treatment allocation, two RCTs did not ensure investigators’ blinding [9,17], which could have been associated with a certain degree of bias. Moreover, two RCTs were underpowered to detect clinical events, with those studies prematurely interrupted during the recruitment phase [12,17]. Very low rates of observed events limited the other two studies during follow-up [13,15]. Other limitations could be associated with those studies recruiting a high proportion of patients with STEMI and multivessel disease [15,16]. The effectiveness of FFR in identifying significant coronary lesions in this setting has been questioned over the years [39,64,65]. Some authors have hypothesized that this technique could underestimate the severity of stenoses, especially in the first days following MI [66]. According to this hypothesis, FFR evaluations of non-culprit lesions following STEMI could be associated with an increased risk of false negatives, leading to incomplete revascularizations and adverse outcomes during follow-up [67,68]. Moreover, FFR evaluations in patients with STEMI and multivessel disease often require multiple coronary procedures, which can also influence clinical outcomes [15].

## 8. The Role of the Defer Groups in RCTs

Besides directing coronary interventions only to those lesions causing ischemia, FFR-guided revascularization benefits are also linked to its capacity to safely defer useless interventions associated with procedural- and therapy-related complications [45]. Therefore, it is reasonable to suppose that the rate of patients undergoing a PCI or CABG in the FFR arm and the ratio between these patients and those undergoing revascularization in the angiography arm are somehow important in determining the relative benefit of an FFR-guided strategy. Deferring can take one of two different types, one involving lesions and one involving procedures. If FFR leads to reduced revascularized lesions, a lower rate of stent/graft failure-related complications can be expected. On the other hand, if FFR leads to a reduced number of performed procedures, along with a reduced rate of device failures, procedural/postprocedural-related complications could potentially be reduced. At the lesion level, both RCTs and non-randomized studies have shown that FFR is highly effective in deferring coronary intervention and effectively reduces the number of useless stents or surgical grafts positioned [7,8,9,14,15,20,21,22]. On the other hand, results from RCTs and observational studies are not as concordant regarding the rates of deferred procedures in the FFR arms. In retrospective studies, the rates of patients undergoing a PCI are mostly between 30% (sometimes even less) and 50% [4,20,23,24,25,27] and are often significantly lower compared to those in the angiography arms. Comparatively, the rates of performed PCIs in the FFR arms of the reported RCTs are mostly between 70% and 100% [7,8,10,12,15,16], and only a few of these studies demonstrated a significant reduction in PCIs compared to the angiography arm [9,10,12,16]. The features of the study design could partially explain this discrepancy in RCTs. Considering the reported studies, five RCTs recruited patients with multivessel coronary disease (who were at high risk of being treated with a PCI or CABG regardless of the treatment allocation) [7,13,14,15,16], and two RCTs recruited patients who all underwent a PCI before randomization [8,15]. Besides the role played by the study design in determining the number of patients undergoing a PCI or CABG in a certain RCT, specific angiographic inclusion criteria could further influence the outcomes of these studies. An FFR evaluation should be recommended for patients with angiographically intermediate lesions [6]. Indeed, performing invasive physiological evaluations on extremely tight lesions (ideally >90%) can be associated with futility; on the other hand, it is highly unlikely that performing these on poorly diseased coronary arteries could be associated with improved diagnostic performance. Moreover, it has been demonstrated that the systematic use of pressure guidewires can lead to an increased risk of iatrogenic coronary dissections [17]. Overall, using FFR in non-intermediate lesions can play an important role in determining the rates of deferred patients in the angiographic or FFR arms, strongly influencing the final results of a certain RCT. Despite the correlation between visual estimation and FFR being generally poor, it has been demonstrated that an angiographic threshold near 50% of stenosis diameter is associated with the highest diagnostic accuracy for ischemic FFR values in most patients [69]. Moreover, these lesions should be further investigated to identify MI effectively. Despite these considerations, angiographic inclusion criteria dramatically differed across the reported RCTs, ranging from true moderate lesions [11] to the systematic use of FFR in all coronary arteries, regardless of the angiographic stenosis diameter [17]. Additionally, it should be noted that several RCTs investigating FFR-guided revascularization included a high proportion of patients with ACS [7,10,12,15,16]. If we do accept the hypothesis that intervention deferral plays an important role in defining the final results of a certain study, including a high number of acute patients with an elevated risk of undergoing coronary revascularization could partially reduce the detectable benefits of FFR. Overall, most of these considerations can easily justify why deferred intervention groups were so underrepresented in the cited RCTs compared to the ones reported in observational studies.

## 9. Conclusions

Several studies have demonstrated that FFR-guided revascularization is safe and highly effective in selecting lesions needing revascularization and deferring those which do not. However, data from RCTs comparing FFR and angiography led to conflicting results. Although these discordant findings can be partially related to specific RCT designs and limitations, no definitive conclusions can currently be drawn. Recently, the results of the “FFR Versus Angiography-Guided Strategy for Management of AMI With Multivessel Disease” (FRAME-AMI, NCT02715518) trial have been presented at the annual congress of the European Society of Cardiology (ESC). According to the authors’ presentation, FFR-guided revascularization was associated with improved clinical outcomes (including hard endpoint as death and MI) when compared to angiography in patients with ACS and multivessel disease undergoing complete revascularization of non-culprit lesions. More details will be available following the publishing of this study.

Recently, the “International Study of Comparative Health Effectiveness With Medical and Invasive Approaches” (ISCHEMIA) trial questioned the benefits of coronary revascularization over medical therapy in terms of hard endpoint reduction in patients with stable coronary artery disease [59,70]. In particular, coronary revascularization was associated with an increased risk of peri-procedural MI and reduced rates of spontaneous MI during follow-up without influencing the final outcomes. According to these findings, patients eligible for coronary revascularization should be selected with even more attention paid to reducing the risk of futility and peri-procedural complications and performing interventions only on patients who can benefit from them. The “Fractional Flow Reserve (FFR) Guided Percutaneous Coronary Intervention (PCI) Plus Optimal Medical Treatment (OMT) Verses OMT” (FAME 2) trial compared FFR-guided revascularization to medical therapy in 1220 patients with stable coronary artery disease. This strategy was associated with a significant reduction in the composite endpoint (4.3% vs. 12.7%, *p* < 0.001), primarily due to the reduction in further urgent revascularizations and MI [71]. These findings were confirmed even at a long-term follow-up (median: 35 months) in a recent patients-pooled meta-analysis, including data from the FAME 2, the Compare Acute, and the “Primary PCI in Patients With ST-elevation Myocardial Infarction and Multivessel Disease: Treatment of Culprit Lesion Only or Complete Revascularization” (DANAMI-3-PRIMULTI) trials (cardiac death or MI: 10.7% vs. 16.4%, *p* = 0.04) [72]. Overall, FFR could potentially boost the benefits of coronary revascularization by correctly identifying coronary lesions that need revascularization and safely deferring those that do not. Future large RCTs and patient-level meta-analyses are required to address this issue and help physicians who perform daily coronary interventions identify those patients who can experience improved outcomes following coronary revascularization.

## Figures and Tables

**Table 1 diagnostics-12-02659-t001:** RCTs investigating FFR-guided vs. angiography-guided coronary revascularization.

Study and Year	Country	Patients (FFR vs. Angio)	FFR Threshold	Coronary Setting	Clinical Endpoints	Follow-Up (Months)
FAMOUS N-STEMI	Scotland	176 vs. 174	0.80	NSTEMI	MACE, death, MI,	12
DK-CRUSH VI	China	160 vs. 160	0.80	Bifurcation lesions	MACE, death, MI, TVR	12
DEFER DES	South Korea	114 vs. 115	0.75	MVD and CCS or ACS	MACE, death, MI, TLR, any revascularization	60
FAME	Netherlands, Belgium	509 vs. 496	0.80	MVD and CCS or ACS	MACE, death, MI, any revascularization	60
Zhang et al., 2016 [10]	China	110 vs. 110	0.80	NSTEMI	MACE, death, MI	12
Thuesen et al., 2018 [13]	Denmark	49 vs. 48	0.80	CCS or NSTEMI treated with CABG	MACCE, death, MI, any revascularization	6
GRAFFITI	International	88 vs. 84	0.80	CCS or NSTEMI treated with CABG	MACCE, death, MI, TVR	12
FLOWER-MI	France	586 vs. 577	0.80	MVD and STEMI	MACE, death, MI, TLR, any revascularization	12
FUTURE	France	460 vs. 467	0.80	MVD and CCS/ACS	MACCE, death, MI, stroke, unplanned revascularization	12
RIPCORD 2	UK	552 vs. 548	0.80	CCS, NSTEMI	MACCE, death, MI, unplanned revascularization	12

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
