# Peer review of "Fractional Flow Reserve-Guided Coronary Revascularization: Evidence from Randomized and Non-Randomized Studies"

_diagnostics, 2022, doi:10.3390/diagnostics12112659_

Round 1

Reviewer 1 Report

This is a very well written article on FFR in coronary artery disease describing findings from RCT and non-RCT trials.

Introduction, rationale, metod of FFR assessment, evidence are well prepared and clear.

My only one remark is that Authors could address short paragraph on FFR in special circumstances like aortic stenosis or MINOCA 

Author Response

We do thank the reviewer for his comments

Esxtensive description of the role of FFR in patients with INOCA/MINOCA was beyond the purposes of our paper. However, we do recognize that this is a topic of interest for those who may read our work.

Therefore, a specific statement has been provided at the end of the paragraph regarding FFR principles

Reviewer 2 Report

There is not specific comments reltaed to this article.

Author Response

We do thank the reviewer for his comments.

Some corrections to the English have been perfomed and properly tracked in the text.

Reviewer 3 Report

I appreciate this work as important and necessary.

Author Response

We do thank the reviewer for his comments